# Impact of Size and Distribution of k-Carbides on the Hydrogen Embrittlement and Trapping Behaviors of a Fe-Mn-Al-C Low-Density Steel

**DOI:** 10.3390/ma17112698

**Published:** 2024-06-03

**Authors:** Yinchen Xiong, Xiaofei Guo, Han Dong

**Affiliations:** 1School of Materials Science and Engineering, Shanghai University, Shanghai 200444, China; xiongyinchenshu@gmail.com; 2State Key Laboratory of Advanced Special Steel, Shanghai University, Shanghai 200444, China

**Keywords:** Fe-Mn-Al-C low density steel, k-carbides, hydrogen embrittlement

## Abstract

This study compares the hydrogen embrittlement susceptibility of a Fe-30Mn-8Al-1.2C austenitic low-density steel aged at 600 °C for 0 (RX), 1 min (A1) and 60 min (A60), each exhibiting varying sizes and distributions of nano-sized κ-carbides. Slow strain rate tests were conducted to assess hydrogen embrittlement susceptibility, while thermal desorption analysis was applied to investigate hydrogen trapping behaviors. Fracture surface analysis was employed to discuss the associated failure mechanisms. The results suggest that nano-sized κ-carbides with sizes ranging from 2–4 nm play a crucial role in mitigating hydrogen embrittlement, contrasting with the exacerbating effect of coarse grain boundary κ-carbides. This highlights the significance of controlling the sizes and morphology of precipitates in designing hydrogen-resistant materials.

## 1. Introduction

Fe-Mn-Al-C austenitic low-density steels have gained increasing attention as potential lightweight steels due to their exceptional mechanical properties, formability and high specific strength (ratio of strength to mass density) [1,2,3]. These steels typically contain 12–30% Mn, 5–12% Al, and 0.6–2.0% C, offering a significant weight reduction potential and maintaining a stable austenitic phase at room temperature. They exhibit tensile strengths ranging from 600–1100 MPa and a ductility ranging from 40–100% [3,4,5]. During the aging process, κ-carbides with a perfect structure L12′ ((Fe, Mn)_3_AlCx (x < 1)) tend to precipitate. These carbides maintain a cube-on-cube coherency with the parent austenitic phase, with the lattice mismatch at the interface typically around 2%, increasing the aging time [6,7,8,9,10,11,12]. The κ-carbides primarily precipitate through spinodal decomposition during aging and are uniformly distributed in the matrix [9,10,13,14,15,16]. These nano-precipitates effectively enhance the yield strength without significantly sacrificing plasticity, resulting in a high yield-to-tensile strength ratio of 0.84–0.95 [16,17,18,19,20,21].

Elkot et al. recently reported a significant increase in yield strength from 540 MPa to 875 MPa in a Fe-28.4Mn-8.3Al-1.3C (wt.%) steel while maintaining a high ductility ranging from 43–73% after aging at 550 °C for 0, 1 and 16 h, respectively [22]. With prolonged aging, κ-carbides tend to form along grain boundaries (GBs), rapidly coarsening into disc-shaped structures and decorating the GBs continuously [23,24]. This results in a notable reduction in ductility due to GB decohency. Sun et al. [25] noted that the formation of GB κ-carbides is faciliated by the contact of GB with precursor-state κ-carbides formed through spinodal decomposition. Conseqently, GB κ-carbides form much rapidly. Liu et al. [26] observed the appearance of sub-micro unshearable κ-carbides at GBs, leading to stress concentration and reduced ductility. Therefore, controlling the size, fraction and distribution of κ-carbides is crucial for improving the mechanical properties of austenitic low-density steels.

Hydrogen embrittlement (HE) is a degradation phenomemon characterized by a significant reduction in mechanical properties when internal or environmental hydrogen is present [27,28,29]. Koyama et al. [30] observed a remarkable decrease of ductility from 44% to 5% in a Fe-26Mn-11Al-1.2C (wt.%) austenitic low-density steel with only 0.14 ± 0.05 ppm diffusive hydrogen. The embrittlement mechanisms were primarily attributed to GB decohesion at triple junctions and interactions with micro-bands. In the mentioned work, the investigated material exhibited a high yield strength of 1009 ± 15 MPa, indicating significant coarsening of κ-carbides. As mentioned in the work from Elkot et al. [22], with GB κ-carbides sized 13 nm, hydrogen-enhanced decohesion (HEDE) was more likely to occur at κ-carbide/matrix interfaces at GBs. Timmerscheidt et al. [31] calculated in accordance with the density function theory that hydrogen atoms tend to be trapped by carbon vacancies of κ-carbides through the formation of M-H bonds. The effects of κ-carbides with a finer size or under short-range-ordering conditions remain to be explored.

Wei and Tsuzaki et al. [32] investigated a 0.05C-0.20Ti-2.0Ni steel processed with iced-brine quenching (IBQ) to produce varying sizes of incoherent TiC carbides. They found that HE sensitivity increased as TiC precipitates transitioned from semi-coherent to non-coherent with the matrix during coarsening. The hydrogen trapping energy increased as the coherency decreased. Additionally, κ-carbides interacted with dislocations, forming high-density dislocation walls (HDDWs) and facilitating glide plane refinement [5,33,34,35,36]. In an austenitic (CoCrNi)94Al3Ti3 (at.%) medium entropy alloy with L12′ (γ’ phase) precipitates ranging in size from 20 to 140 nm, Cheng et al. [37] revealed that the trapping energy of H-dislocation interactions was around 25 kJ/mol, smaller than the 30 kJ/mol from H-L12′ precipitates. This led to a dynamic relocation of hydrogen atoms between mobile dislocations and precipitates when dislocations cut though precipitates. Kim et al. [38] demonstrated in a CrMnFeCoNi high-entropy alloy that hydrogen distorted intertial sites, retarded dislocation gliding, and reduced stacking fault energy. Thus, the local deformation mechanisms were affected. Changes in the local deformation mechanism due to hydrogen and κ-carbides precipitation can affect HE sensitivity.

Hence, this study compares the HE subceptibility of a Fe-30Mn-8Al-1.2C austenitic low-density steel after being aged at 600 °C for 0 (RX), 1 min (A1) and 60 min (A60), exhibiting varied sizes and distributions of nano-sized κ-carbides, through a slow strain rate test. Additionally, the hydrogen trapping behavior was investigated using thermal desorption analysis. The failure mechanisms were discussed with fracture surface analysis.

## 2. Materials and Methods

The investigated steel Fe-30Mn-8Al-1.2C has the main chemical composition listed in Table 1. It was produced through strip casting, homogenized at 1150 °C for 5 h and cold-rolled with a total reduction of 50%. Subsequently, it underwent recrystallization annealing in a salt bath at 900 °C for 60 min (RX), followed by water quenching. To obtain varying sizes of κ-carbides, the RX specimens were further aged at 600 °C for 1 min (A1), 60 min (A60) in a salt bath to facilitate κ-carbide precipitation, followed by water quenching.

Diffusive hydrogen was introduced into the specimens via electrochemical charging in 0.05 M H_2_SO_4_ + 1 g/L CH_4_N_2_S at a constant current density of 5 mA/cm^2^ for 24 h. Prior to charging, the specimen surfaces were mechanically ground up to #2000 SiC paper and polished with 6 μm polishing clothes. Following charging, the specimens were ultrasonically cleaned and directly tensile-strained until fracture in a slow strain rate test. The fracture tip region was cut and stored in liquid N_2_ for further hydrogen measurement. Hydrogen content was measured by thermal desorption spectrometry (TDS-System, Bruker Galirio 8, Ettlingen, Germany) at a constant heating rate of 20 °C/min from room temperature to 800 °C, to study hydrogen trapping behavior. Desorption peaks were further analyzed using Origin^®^ software, employing Gaussian peak functions to distinguish diffusive hydrogen (H_diff_) and trapped hydrogen (H_trap_).

The slow strain rate test (SSRT) was conducted using constant extension equipment at a constant strain rate of 10^−6^ s^−1^ at room temperature. Dog-bone-shaped tensile specimens with gauge dimensions of 25 mm × 5 mm × 1 mm were used, in which the gauge length aligned along the rolling direction. The HE susceptibility was assessed using the hydrogen embrittlement index (HEI), which comprises HEI_UTS_ (hydrogen embrittlement index of ultimate tensile strength) and HEI_EL_ (hydrogen embrittlement index of uniform elongation). These indices represent the reduction in ultimate tensile strength and uniform elongation compared to the uncharged conditions.

The microstructural fracture surfaces were characterized using a scanning electron microscope (SEM, ZEISS Sigma 300, Oberkochen, Germany) equipped with electron backscatter diffraction (EBSD). Specimens for EBSD observations were polished with 0.02 μm colloidal silica (OPS) for approximately 20 min. EBSD analysis was conducted with a step size of 0.1 μm and analyzed using Aztec Crystal software. The precipitation behavior of κ-carbides and the microstructure near the fracture tip region were further examined using transmission electron microscopy (Thermofisher Talos F200X, Waltham, MA, USA).

## 3. Results

### 3.1. Microstructure and Mechanical Properties

The microstructure of the RX state was analyzed using SEM and EBSD, as shown in Figure 1. It reveals a fully recrystallized face-centered cubic microstructure with annealing twins. The average grain size is approximately 22 ± 5 μm. The grain size was measured on SEM microstructure images according to the linear intercept method using the software Image J^®^.

The aging treatment at 600 °C did not affect the grain morphology but influenced the size and distribution of κ-carbides. Figure 2 presents TEM bright-field images of the microstructure after different aging periods. In the RX state (Figure 2a), both inter- and intra-kappa phases were undetectable, although a weak selected area electron diffraction (SAED) pattern of κ-carbides was observed. The presence of excessive Mn, Al elements likely led to the formation of chemical clusters or short-range ordering (SRO). In the A1 annealed state (Figure 2b), κ-carbides emerged clearly after 1 min annealing. Dark-field images show κ-carbides with sizes ranging from 2–4 nm dispersed uniformly within the grain. The size of κ-carbides was determined by performing an Inverse-Fast Fourier transform (IFFT) of the Fast Fourier transform (FFT) pattern of k-carbides from high-angle annular dark-field (HAADF) images. An FFT analysis revealed a distinct diffraction pattern of L’12, indicative of their globular morphology. Additionally, grain boundary κ-carbides exhibited a similar size and morphology to the intra-granular ones. In the A60 annealed condition (Figure 2c), intra-granular κ-carbides coarsened to sizes ranging from 7–9 nm, taking on a cuboidal morphology. FFT analysis confirmed the presence of strong diffraction patterns of L’12, covering both grains in the grain boundary region. The intra-granular k-carbides revealed a cuboidal morphology. At the grain boundaries, larger κ-carbides were observed, suggesting accelerated coarsening in this region.

### 3.2. Hydrogen Resistance Evaluated by Slow Strain Rate Test

Slow strain rate tests (SSRTs) were conducted on both uncharged and 24 h H pre-charged specimens to compare their hydrogen embrittlement susceptibility after different aging treatments. Figure 3 illustrates the engineering stress–strain curves of these specimens. Notably, specimens from all three aging conditions exhibited reductions in both tensile strength and ductility. For the RX specimens, the ultimate tensile strength (UTS) decreased from 887 MPa to 771 MPa, corresponding to a HEI_UTS_ of 13.1%. Similarly, the uniform elongation (UE) declined from 77.3% to 63.4%, resulting in a HEI_EL_ of 18.0%. In comparison, the uncharged A1 specimens showed a UTS and EL of 891 MPa and 64.1%, which declined to 857 MPa and 54.3%. The HEI_UTS_ and HEI_EL_ are equivalent to 3.8% and 15.4%, respectively. For the A60 specimens with coarsened k-carbides, the UTS was 960 MPa in the uncharged condition, reducing to 919 MPa, while the EL decreased from 22.1% to 3.7%. The HEI_UTS_ and HEI_EL_ were equivalent to 4.3% and 83.2%, respectively. This demonstrated a clear trend showing that the HE susceptibility increased with longer aging periods and larger sizes of k-carbides, as evidenced by the significant reduction in elongation.

### 3.3. Thermal Desorption Analysis

To investigate the hydrogen trapping behaviors in the specimens with varying sizes and morphologies of k-carbides, thermal desorption analysis (TDS) was performed on both uncharged and 24 h H pre-charged and SSRT-tested specimens at a constant heating rate of 20 °C/min from room temperature to 800 °C. Figure 4 presents the TDS curves. The Gaussian fitting was used to determine hydrogen content under different trapping peaks.

In Figure 4a, the RX specimen initially contained 4.5 ppm trapped hydrogen, which is the integral of hydrogen under the desorption peak in the uncharged condition (black curve). After 24 h H pre-charging and SSRT for 241 h until failure, it had 9.1 ppm diffusive hydrogen and 4.3 ppm trapped hydrogen. The Al specimen initially had 5.1 ppm trapped hydrogen. After 24 h H pre-charging and SSRT for 216 h until failure, it had 17.2 ppm diffusive hydrogen and 6.4 ppm trapped hydrogen, as shown in Figure 4b. In contrast, the A60 specimen contained 2.3 ppm trapped hydrogen in the uncharged condition. After 24 h H pre-charging and SSRT testing for only 30 h until failure, this increased to 10.2 ppm diffusive and 2.8 ppm trapped hydrogen. Figure 4d compares the TDS curves from the pre-charged SSRT specimens. The A1 specimen exhibited high levels of trapped hydrogen in the as-aged condition and high diffusive hydrogen after a long SSRT testing period. Specifically, the amount of diffusive hydrogen was 1.7 times that of the A60 specimen, while the trapped hydrogen was 2.3 times higher than that of the A60 specimen. It is observed that the diffusive hydrogen in the A60 specimen started to desorb at lower temperatures compared to the RX specimen, indicating that the diffusible hydrogen in the A60 specimen was more readily released.

### 3.4. Fracture Surface Analysis

Figure 5 presents the fracture surface of SSRT-tested specimens under uncharged (Figure 5a–c) and 24 h H pre-charged (Figure 5d–f) conditions. In the absence of diffusive hydrogen, both the RX and A1 specimens (Figure 5a,b) exhibit uniformly distributed ultrafine yet deep dimples across the fracture surfaces, corresponding to their high elongation values. In contrast, the A60 specimen (Figure 5c) displays a combination of ductile dimples and flat brittle facets, with sizes comparable to the grain sizes. Additionally, slip traces are evident on the brittle facets. Notably, ridges with voids are observed at the curvature of the brittle facets, indicating voids formation at the ridges concurrent with the formation of slip bands during continuous tensile deformation.

Figure 5d presents the fracture surface of the hydrogen pre-charged RX specimen. The central region of the fracture surface exhibits ductile dimples, indicative of significant plastic deformation, despite the presence of 9.1 ppm diffusive hydrogen. A pronounced HE layer, with a depth from 38 to 47 μm, is observed. Brittle facets are discernible along the periphery of the fracture surface, attributed to the initiation of surface cracks during SSRT. In Figure 5e, the hydrogen pre-charged A1 specimen exhibits a shallow hydrogen-affected zone characterized by inter-granular fracture, with a penetration depth ranging from 30 to 35 μm. Despite this, the inner fracture surfaces display uniformly distributed dimples resembling the uncharged condition.

Meanwhile, the fracture behavior of the A60 sample (Figure 5f) reveals a significantly expanded brittle region spanning from 60 to 130 μm. Notably, triple junction cracking and the presence of slip traces on the inter-granular facets are conspicuous. It is postulated that hydrogen diffusion and crack propagation are facilitated in specimens subjected to longer aging periods. Additionally, the inner region shows a mixed ductile and brittle morphology, consistent with its high HEI_EL_ values.

The TDS results reveal that the A60 specimen exhibited the lowest levels of both diffusive and trapped hydrogen compared to the RX and A1 specimens. However, it also displayed a notably higher depth of the brittle surface layer. This pronounced HE layer can be attributed to the promoted hydrogen-enhanced decohesion of grain boundaries. This effect is not solely due to diffusive hydrogen, but also to the sensitive microstructure features of the A60 specimen.

## 4. Discussion

Both the SSRT results and the fracture surface analysis indicate that the A1 specimen exhibited the lowest HEI_UTS_ of 3.8% compared to RX and A60 specimens. The HEI_EL_ of A1 is 15.3%, which is also lower than those from RX (18.0%) and A60 (83.2%) specimens. The reduction in elongation per unit for hydrogen would be much less considering the A1 specimen had 17.2 ppm diffusive hydrogen. As compared with the HE susceptibility reported in Fe-26Mn-11Al-1.2C steel by Koyama et al. [30], both the RX and A1 specimens reveal a better HE performance. The A1 specimen features nanoscale κ precipitates with a size between 2–4 nm with a globular-shaped morphology. These precipitates demonstrated a high capability in raising the yield strength, reducing the HEI and promoting a ductile-dominated failure mode. Additionally, the A1 specimen exhibits a significantly higher amount of reversible-trapped (diffusive) hydrogen, as observed from TDS measurement, compared to other aged conditions. This underscores the beneficial role of nano-sized κ-carbides as effective hydrogen traps and in mitigating hydrogen embrittlement.

The presence of a thin brittle layer on the fracture surface analysis, as depicted in Figure 5e, also suggests that the high trappability of diffusive hydrogen prevents hydrogen ingression. As proposed by Timmerscheidt et al. [31], hydrogen tends to be trapped by the carbon vacancies on the interface of k-carbides. Given that the A1 specimen has nano-sized 2–4 nm k-carbides with a globular shape possessing higher surface areas compared to other irregular shapes, this would inevitability enhance hydrogen trapping.

The A60 specimens exhibit a mixed fracture mode in the absence of hydrogen. The fracture morphology suggests that grain boundary decohesion occurred. TEM micrographs from the fracture tip region in the H pre-charged and SSRT-tested A60 specimen, which failed at a uniform elongation of 3.7%, are presented in Figure 6. In Figure 6a, dense micro-bands, spaced at 150–300 nm intervals, are observed at this low deformation level. The intersection of micro-bands with the grain boundaries results in the accumulation of dislocations and stress concentrations, which were proposed as reasons for crack nucleation [24,25,26,30]. Additionally, Figure 6b reveals the presence of coarse grain boundary k-carbides. These carbides are densely distributed along the GBs, with a size above 100 nm. As references, the red circles have a diameter of 150 nm. It is believed that these large GB k-carbides reduce grain boundary cohesion force due to their high coherency misfit with the matrix [22,23]. Consequently, voids and cracks are more likely to form at these sites, leading to an expanded cracking behavior in the presence of hydrogen.

## 5. Conclusions

This study investigated the microstructure features and hydrogen embrittlement (HE) behavior of a Fe-30Mn-8Al-1.2C low-density steel under recrystallization annealed conditions (RX) and aged at 600 °C for 1 min (A1) and 60 min (A60). The main conclusions are as follows:The material shows a fully recrystallized face-centered cubic microstructure with annealing twins and an average grain size of about 22 ± 5 μm. Aging promotes the precipitation of k-carbides in A1 specimens (2–4 nm, globular-shaped), as well as GB k-carbides of a similar size and shape. Conversely, A60 specimens exhibit coarsened intra-granular κ-carbides (7–9 nm, cuboidal-shaped) and coarse GB κ-carbides (above 100 nm), indicating faster coarsening at the grain boundary region.SSRT testing reveals an increasing HE susceptibility with aging time and larger κ-carbide sizes, reflected in a significant elongation reduction. A60 specimens display a mixed fracture mode with grain boundary decohesion, likely due to coarse grain boundary κ-carbides, reducing grain boundary cohesion and leading to the exacerbation of cracking behavior in the presence of hydrogen.TDS reveals higher diffusive and trapped hydrogen levels in the A1 specimens compared to RX and A60 specimens. The high trappability of diffusive hydrogen in A1 specimens is attributed to nano-sized globular κ-carbides. Conversely, A60 specimens exhibit the lowest diffusive and trapped hydrogen levels but a higher surface layer brittleness, suggesting an enhanced hydrogen embrittlement susceptibility due to microstructural sensitivity.

Overall, nano-sized κ-carbides play a crucial role in mitigating hydrogen embrittlement, while the coarse κ-carbides exacerbate it, emphasizing the importance of microstructural control in designing hydrogen-resistant materials.

## Figures and Tables

**Figure 1 materials-17-02698-f001:**
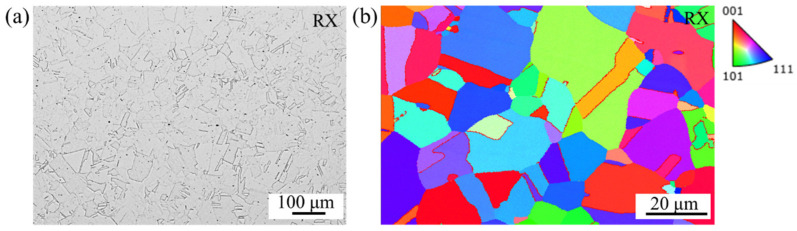
Microstructure of the investigated Fe-30Mn-8Al-1.2C steel in recrystallization annealed (RX) condition: (**a**) SEM micrograph; (**b**) EBSD inverse pole figure.

**Figure 2 materials-17-02698-f002:**
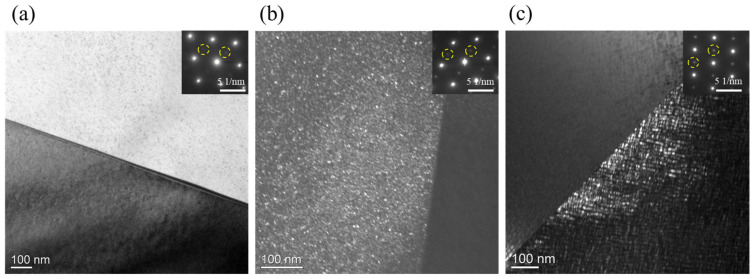
TEM micrographs of the investigated Fe-30Mn-8Al-1.2C steel: (**a**) precipitates near the grain boundary region in RX states, the SAED pattern of intra-granular k-carbides is inserted; (**b**) precipitates near the grain boundary region in A1 states, the SAED pattern of intra-granular k-carbides is inserted; (**c**) precipitates near the grain boundary region in A60 states, the SAED pattern of intra-granular k-carbides is inserted, the superlattice reflections indicates the chemically ordered k-carbides were highlighted by yellow circles.

**Figure 3 materials-17-02698-f003:**
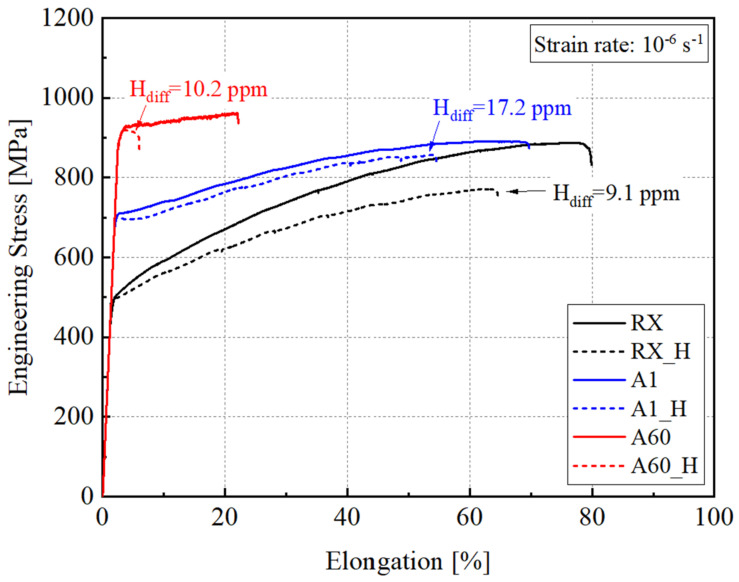
Engineering stress–strain curves of the investigated Fe-30Mn-8Al-1.2C steel in RX, A1 and A60 aged states with and without 24 h H pre-charging (the slow strain rate of 10^−6^ s^−1^ was applied).

**Figure 4 materials-17-02698-f004:**
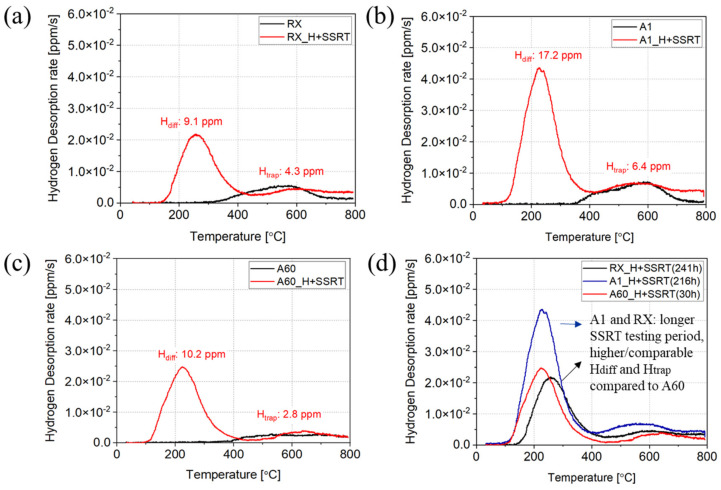
Thermal desorption spectra from (**a**) RX, (**b**) A1, (**c**) A60 specimens in the H uncharged (black curves), 24 h H pre-charged and SSRT-tested conditions (red curves); (**d**) comparison of the 24 h H pre-charged and SSRT-tested RX, A1 and A60 (the time in the parentheses indicates the time to failure in SSRT) (the heating rate is 20 °C/min, from RT to 800 °C).

**Figure 5 materials-17-02698-f005:**
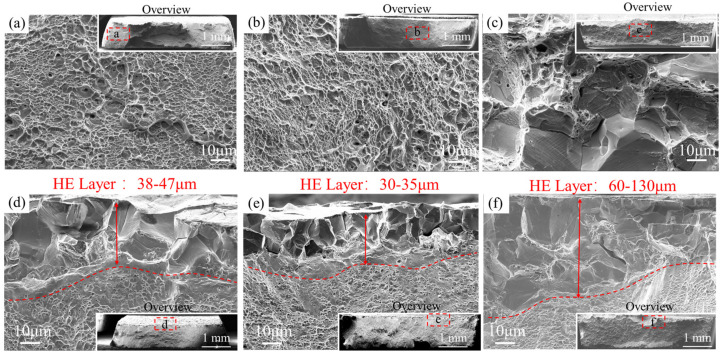
Fracture surfaces of (**a**–**c**) uncharged and SSRT-tested RX, A1 and A60; (**d**–**f**) 24 h H pre-charged and SSRT-tested RX, A1 and A60; the overviews of the fracture surfaces were inserted in the corner of each image with indicating the image taken location in red square.

**Figure 6 materials-17-02698-f006:**
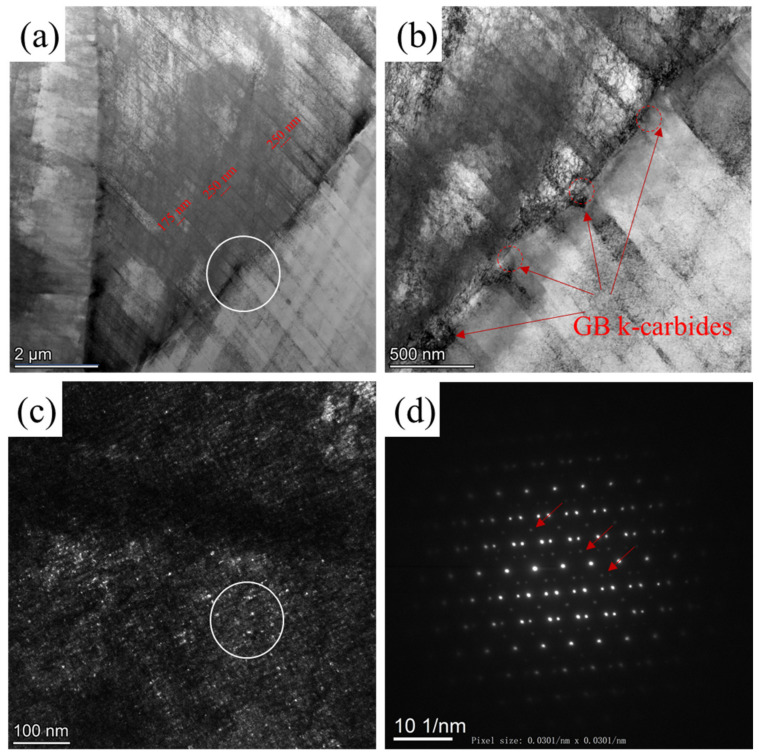
TEM micrographs taken from the fracture tip region in the H pre-charged and SSR-tested A60 specimen: (**a**) bright-field image revealing the micro-bands and grain boundary interactions; (**b**) bright-field image with grain boundary k-carbides and the associated stress concentration; (**c**) dark-field image from (**a**), and the associated (**d**) SAED pattern of the circled region in (**c**); the red arrows indicate the superlattice reflection of k-carbides.

**Table 1 materials-17-02698-t001:** Main chemical composition of the investigated Fe-30Mn-8Al-1.2C steel.

Elements	Fe	C	Si	Mn	S	Al
wt.%	balanced	1.27	0.09	29.65	0.02	8.38

## Data Availability

The original contributions presented in the study are included in the article, and further inquiries can be directed to the corresponding authors.

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
