# Peer review of "Impact of Size and Distribution of k-Carbides on the Hydrogen Embrittlement and Trapping Behaviors of a Fe-Mn-Al-C Low-Density Steel"

_materials, 2024, doi:10.3390/ma17112698_

Round 1

Reviewer 1 Report

Comments and Suggestions for Authors

The introduction describes the problem well, the aim is to study the effect of annealing on hydrogen embrittlement, and to study the effect of the size and distribution of the kappa phase on the ability to capture hydrogen.

The methodology describes in some detail the process of obtaining steel samples. The Slow strain rate test (SSRT) is usually performed in some environment (various solutions), but according to the methodology in this research the steel was saturated with hydrogen before the test. In this case it is not clear why the thermal desorption spectra were taken from samples after SSRT and not after hydrogen saturation. (Possibly due to the influence of dislocations in hydrogen capture, which was described in the introduction)

In Figure 3, the strain curves include the notations H(diff), up to this point in the paper these notations were not present. Figure 4 (a), possibly a typo is present, the designation of the smaller peak on the spectrum is different from the rest of the spectra (b, c). Information about SSRT time is present, it is not clear what it means (time to fracture or set time). The measurement error of Hydrogen desorption rate is missing.

Figure 5, it is desirable to add to the overview photo the designation of the section where the fracture surface was taken.

The conclusions are sufficiently complete and correspond to the set aim.

Reviewer 2 Report

Comments and Suggestions for Authors

The authors present an investigation on the microstructure features and hydrogen embrittlement behavior of a Fe-30Mn-8Al-1.2C low-density steel under the recrystallization annealed and aged condition at 600°C for 1 min and 60 min. They analyze the sizes and distributions of nano-sized k-carbides, strain rate, thermal desorption, and fracture surface.

The issue of research presented in this study is within the scope covered by materials. The submitted manuscript reports interesting technical developments and scientifically relevant measurements.

Some aspects are unclear, and the manuscript should be revised before publication. Open points to be clarified include:

-       Abstract.  “HE” should be defined.

-       Page 3, line 119. The authors must state, how they obtained the average grain size.

-        Page 3, line127. The authors state “The presence of excessive Mn, Al elements likely led to the formation of 127 chemical clusters or short-range ordering (SRO).” How do they know that”. The authors should be giving some evidence of that.

-       Page 3, lines 129-130. “Dark-filed images show κ-carbides with sizes ranging from 2-4 nm dispersed uniformly within the grain.”. Authors should mention how the determined the size.

-       Page 4, line 134. “… sizes ranging from 7-9 nm ..”. The same comment above.

-       Page 5, figure 3. H dif should be explained in the text when the authors describe figure 3.

-       Page 5, lines 170, 172 and 174. The authors state”… specimen initially contained 4.5 ppm trapped hydrogen.” It. Should be important that the authors describe in the manuscript how they measure the initial contained trapped hydrogen.

-       Page 7, figure 5. To compare the fracture surfaces of the samples, they should have the same magnification, 20um or 10um.

-       Page 7, line 246. The authors state “ … space at 150-200 nm”. They should be described as obtained that value from the figure 6.a, it is difficult to conclude that.

-       Page 7, line 250. “.. above 100 nm”. The same comment above.

-       “SAED” should be define and describe in the manuscript. That word appears in various part of the manuscript without being defined.

Reviewer 3 Report

Comments and Suggestions for Authors

The article titled "Impact of size and distribution of k-carbides on the hydrogen embrittlement and trapping behaviors of a Fe-Mn-Al-C low-density steel" presents several interesting aspects regarding the study of a low-density austenitic steel, Fe-30Mn-8Al-1.2C, and the influence of κ (k-carbides) nanocarbides on its behavior against hydrogen embrittlement (HE).

In the methodology, it indicates:

"The slow strain rate test (SSRT) was conducted using constant extension equipment 100 at a constant strain rate of 10^-6 s^-1 at room temperature. Dog-bone shaped tensile specimens with gauge dimensions of 25 mm × 5 mm × 1 mm were used, with the gauge length aligned along the rolling direction."

I have seen that various standards are sometimes used, including:

ASTM G123 - Standard Practice for Determining Susceptibility of Metallic Materials to Hydrogen Embrittlement in Low-Pressure Gaseous Environments

ISO 11113 - Metallic materials - Testing for hydrogen embrittlement - Slow strain rate technique in gaseous environments.

Do these standards apply to your work?

The study highlights the importance of controlling the size and distribution of κ nanocarbides. It is observed that nanocarbides around 3 nm play a crucial role in mitigating hydrogen embrittlement, in contrast to coarser carbides formed at grain boundaries, which can worsen embrittlement. How can the time factor be related to the distribution of carbides?
